# Feature Analysis of Scanning Point Cloud of Structure and Research on Hole Repair Technology Considering Space-Ground Multi-Source 3D Data Acquisition

**DOI:** 10.3390/s22249627

**Published:** 2022-12-08

**Authors:** Xinming Pu, Shu Gan, Xiping Yuan, Raobo Li

**Affiliations:** 1Faculty of Land Resources Engineering, Kunming University of Science and Technology, Kunming 650093, China; 2Application Engineering Research Center of Spatial Information Surveying and Mapping Technology in Plateau and Mountainous Areas Set by Universities in Yunnan Province, Kunming 650093, China; 3Key Laboratory of Mountain Real Scene Point Cloud Data Processing and Application for Universities in Yunnan Province, West Yunnan University of Applied Sciences, Dali 671006, China

**Keywords:** point cloud hole repair, point cloud feature extraction, point cloud difference analysis, point cloud data acquisition

## Abstract

As one of the best means of obtaining the geometry information of special shaped structures, point cloud data acquisition can be achieved by laser scanning or photogrammetry. However, there are some differences in the quantity, quality, and information type of point clouds obtained by different methods when collecting point clouds of the same structure, due to differences in sensor mechanisms and collection paths. Thus, this study aimed to combine the complementary advantages of multi-source point cloud data and provide the high-quality basic data required for structure measurement and modeling. Specifically, low-altitude photogrammetry technologies such as hand-held laser scanners (HLS), terrestrial laser scanners (TLS), and unmanned aerial systems (UAS) were adopted to collect point cloud data of the same special-shaped structure in different paths. The advantages and disadvantages of different point cloud acquisition methods of special-shaped structures were analyzed from the perspective of the point cloud acquisition mechanism of different sensors, point cloud data integrity, and single-point geometric characteristics of the point cloud. Additionally, a point cloud void repair technology based on the TLS point cloud was proposed according to the analysis results. Under the premise of unifying the spatial position relationship of the three point clouds, the M3C2 distance algorithm was performed to extract the point clouds with significant spatial position differences in the same area of the structure from the three point clouds. Meanwhile, the single-point geometric feature differences of the multi-source point cloud in the area with the same neighborhood radius was calculated. With the kernel density distribution of the feature difference, the feature points filtered from the HLS point cloud and the TLS point cloud were fused to enrich the number of feature points in the TLS point cloud. In addition, the TLS point cloud voids were located by raster projection, and the point clouds within the void range were extracted, or the closest points were retrieved from the other two heterologous point clouds, to repair the top surface and façade voids of the TLS point cloud. Finally, high-quality basic point cloud data of the special-shaped structure were generated.

## 1. Introduction

In recent years, the accuracy of high-spatial-resolution optical sensors and multi-line lidar sensors has gradually improved, manufacturing costs have been declining, and these sensors have been widely used in the fields of automatic driving, forestry investigation, and digital twins. Among them, the massive 3D point cloud data obtained by a sensor expressing the surface characteristics of the target has become indispensable first-hand data for real-world data virtualization. Moreover, the comparative analysis of multi-source point cloud data has been a research hotspot. At present, the acquisition of 3D point cloud data mainly relies on photogrammetry technology and laser scanning technology. However, the spatial entities are represented in the form of three-dimensional coordinates no matter how the 3D point cloud is generated. The data are similar, and the conversion, storage, and processing of the results are consistent [1]. Since two-point cloud acquisition methods have different technical characteristics, there are some differences in data acquisition conditions, acquisition efficiency, and measurement accuracy [2]. Nonetheless, researchers have verified that the above two technologies were used for the same target simultaneously. High-precision feature point clouds containing texture information can be quickly obtained [3]. In this way, the fusion of the two-point cloud data can complement each other to obtain a higher-quality target point cloud. Additionally, a single-point cloud acquisition method may not be suitable for some specific scanning environments, resulting in blind area scanning and point cloud missing. At this time, the combination of multiple data collection methods can improve the integrity of point cloud data. Guo Yu [4] combined TLS scanning with multi-baseline close-up photogrammetry in the cave point cloud acquisition where the laser scanner cannot be normally deployed, and employed multi-source data to complete the repair of the holes existing in the laser point cloud. Haoyu Tai et al. [5] adopted an RGB-D camera and TLS scanner to collect the point cloud of mine maintenance roadway, grappling with the difficulty of missing point cloud data caused by the limited field of view of TLS under too narrow a space.

With the continuous advancement of projects such as Real Scene 3D China and cultural relics digitization, the demand for refined 3D model construction has increased dramatically. At present, the more mature 3D modeling methods are mainly divided into two types: low-altitude photogrammetry image data modeling based on UAS (unmanned aerial system) and laser point cloud modeling. However, the former is affected by many factors such as heading overlap, side-direction overlap, route height, and weather conditions. As a result, it is difficult to guarantee the quality of the final obtained data, and the efficiency of model construction is reduced. In other words, a single modeling method cannot fully meet the requirements of complex 3D model construction. El-Hakim et al. [6] studied the integration of multi-source sensors and believed that the laser point cloud obtained by the laser scanner has the characteristics of high precision, simple operation, and small impact on the external environment, which can complement low-altitude photogrammetry technology. Moreover, the combination of photogrammetry and laser scanning can efficiently obtain detailed models of complex buildings, and the combination of these two methods has completed the three-dimensional reconstruction of Pomposa Monastery. Xie Yunpeng et al. [7] utilized the good low-altitude detection and anti-interference ability of laser point cloud to wrestle with canopy occlusion and eaves occlusion in oblique photogrammetry, and obtained high-precision three-dimensional models of the city under the premise of ensuring the efficiency of three-dimensional modeling through the combination of multi-source data. Through the airborne laser point cloud and image air three registration, the quality of air three can be improved, and the production of a single three-dimensional building model with a clear structure and realistic texture can be realized [8]. Li Peng et al. [1] improved the quality of airborne laser point clouds with image air three registrations and realized the production of a monolithic 3D model with a clear structure and real texture. In the protection of the Tuoshan Grottoes [9], the TLS (terrestrial laser scanner) point cloud, the HLS (hand-held laser scanner) point cloud, and the photogrammetric point cloud were combined through the point cloud registration based on the TLS point cloud, and the multi-source point cloud data was comprehensively used, on the basis of which the refined model construction of the grotto in the complex space environment was realized. Frueh et al. [10] combined two-dimensional images with laser scanning point clouds to complete the reconstruction of the façade grid of Berkeley Downtown Avenue in the presence of obstructions such as sidewalk trees on the façade of the building. Regarding surface areas with a small amount of vegetation, Grebby et al. [11] integrated terrain variables obtained by lidar to significantly enhance the overall lithological mapping accuracy by 22.5% compared to lithological maps obtained by hyperspectral imagery alone. Nonetheless, the point clouds obtained by the lidar carried by different platforms are inevitably defective due to the limitations in the field of view. Hence, the comprehensive utilization of air–ground multi-source point clouds is an inevitable trend. Liu Yakun et al. [12] integrated and complemented the airborne, vehicle-mounted, and ground-based point clouds of the teaching building in the study area through conjugated features, improved the overall quality of the point clouds, and provided good data support for the editing of BIM model attributes. Borkowski et al. [13] fully acquired the point cloud information of the target building by combining ground laser scanning with on-board laser scanning and achieved high-precision semi-automatic modeling with the digital images collected by the camera.

Although the above research has achieved certain results, there is a lack of comparative analysis of the differences and accuracy of the geometric characteristics of the point clouds collected in different ways. In the overlapping regions of the data of different point clouds, most of them are filtered by the same conditions after being merged to reduce the data redundancy of the overlapping areas. Meanwhile, the data quality of different point clouds is not considered, resulting in inevitably incorrect exclusion of the high-quality data in a certain point cloud to a certain extent. Consequently, the integration of point clouds cannot achieve the effect of complementary advantages. It may even result in a decrease in the quality of point clouds in overlapping areas. Therefore, the point cloud acquisition principle of UAS low-altitude photogrammetry technology, handheld 3D laser scanning technology, and ground station 3D laser scanning technology was analyzed in this study. Different schemes were designed according to the technical characteristics to collect the point cloud data of the same structure. The differences in the geometric characteristics of the multi-source point cloud of the same target were compared. Finally, the feature point data and hole data were extracted from the HLS point cloud and UAS point cloud, according to the difference analysis results and the TLS point cloud, to fulfill the repair of the TLS point cloud and achieve the effect of improving the quality of TLS point cloud data.

## 2. Materials and Methods

### 2.1. 3D Data Acquisition Technology and Research Methods

#### 2.1.1. Air–Ground 3D Data Acquisition Technology

In the digital era, 3D point clouds have become the most direct and important 3D spatial data for portraying the real world and can quickly obtain the three-dimensional form of the target [14]. The terrestrial 3D laser scanning technology and low-altitude photogrammetry technology were used to complete the acquisition of the laser point cloud of the structure and the photogrammetry point cloud of the unmanned aerial vehicle so as to obtain the 3D point cloud of the target structure and digitally express it.

According to the different platforms, 3D laser scanning technology can be divided into ground, airborne, satellite-borne, and other types. In this study, two ground-based 3D laser scanners, namely the I-Site8200ER terrestrial 3D laser scanner (TLS, Terrestrial 3D Laser Scanner) produced by MAPTEK Company and the ZEB-HORIZON hand-held 3D laser scanner (HLS, Hand-held 3D Laser Scanner) produced by GeoSLAM Company, were used for laser point cloud acquisition. The parameters of the two instruments are listed in Table 1.

No matter what kind of 3D laser scanner directly obtains the target surface point cloud through the laser, there are still some differences in the principle of data acquisition for different 3D laser scanners. The terrestrial 3D laser scanner I-Site8200ER used in the study transmits the near-infrared wavelength laser to the target surface at a fixed station by the multi-pulse laser emitter in the instrument, and then reflects back to the laser receiver along the same path by the diffuse reflection of the target surface. Next, the target clustering is calculated by the time difference between the emission and receiving laser. The center point position of the laser emitter and the vertical and horizontal angles at observation time are combined to infer the final position of the target point. The hand-held 3D laser scanner ZEB-HORIZON does not need to set a fixed station. A near-infrared laser is launched to the target only by combining the rotation of the motor using a 16-wire lidar. Afterwards, the angular velocity and acceleration of the sensor in space are obtained using the 2D TOF (2D time-of-flight) laser ranging sensor, an inertial measurement unit IMU (inertial measurement unit), and an accelerometer. The position and timestamp information of the sensor relative to the starting point in the measurement process are inferred. The Kalman filter is used to obtain the accurate attitude of the sensor and the relative position relationship of the target point cloud. Subsequently, the cumulative error is reduced by the accurate constraints provided by the loopback detection to acquire the target point cloud. Since HLS data acquisition is more convenient, flexible, small, and lightweight than TLS, HLS data acquisition is more convenient and flexible than TLS. As a result, it has a higher acquisition rate and is suitable for more scenarios, such as small underground spaces, complex roadways, and indoor spaces. The TLS application in the above space will significantly curtail the data acquisition efficiency. In a complex environment, the integrity of data acquisition is difficult to guarantee if the station location cannot be reasonably set. Nevertheless, the TLS point cloud data acquisition method does not have cumulative error in contrast to the HLS point cloud. Hence, it has higher data accuracy.

Low-altitude photogrammetry technology is a platform for unmanned aerial systems (UAS, unmanned aerial system). The high-resolution optical sensors they are with are adopted to obtain high-spatial resolution optical images of the target area through ultra-low-altitude flight. Next, DOMs (digital orthophoto map), DSMs (digital surface model), photogrammetry point clouds, and other basic application data are provided for the surveying and mapping of the area. In this study, the DJI Phantom 4 RTK UAV (unmanned aerial vehicle) and its supporting optical sensor were employed to collect image data. The sensor parameters are listed in Table 2. After the image data acquisition was completed, the image data with a certain degree of overlap is processed by combining SFM (structure from motion) technology and MVS (multi view stereo) technology to generate the UAS photogrammetric point cloud required for the study. This method is utilized to extract the SIFT (scale-invariant feature transform) features in the image and match the image pair according to the geometric constraints of the outer pole, obtain the sensor pose and scene structure, and optimize the pose and scene structure by beam adjustment to complete the sparse reconstruction. Afterwards, stereoscopic correction of any image pair is performed using the camera parameters obtained when the image pair is matched, and all image pairs are aligned to the same plane. Finally, the feature points are matched by limit constraints to generate dense point clouds, that is, UAS photogrammetric point clouds. Bentley Acute3D’s Context Capture software uses the above method to process the image. The UAS photogrammetry point cloud of the research object is obtained through the software.

#### 2.1.2. Comparative Analysis Method of 3D Data Features

In our study, three 3D point cloud data sets were collected: TLS point cloud, HLS point cloud, and UAS photogrammetry point cloud. The single point features of the point cloud were compared and analyzed from the perspective of the geometric domain to fully grasp the data characteristics of different point clouds. In addition, the TLS point cloud was selected as the main reference point cloud. The density and roughness of the TLS point cloud and HLS point cloud were calculated with the same field radius. The calculation results were compared and analyzed. Through the point cloud visualization, the differences in the acquisition effect of TLS point cloud and UAS photogrammetric point clouds in the small component area, the trench-like texture area, and the vegetation occlusion area were analyzed, as well as the large-scale void loss of the cloud top surface and facade at different points.

The target structure of this study is not liable to produce large deformations under the action of no external force conditions. Since the acquisition of different instrument data is completed in a short time interval, the point cloud data obtained by different methods should be the same or extremely similar without considering the system error of the instrument and equipment and the small changes in the external environment. Then, the study performed M3C2 (Multiscale Model-to-Model Cloud Comparison) [15] distance statistics on the three-point cloud data from the perspective of single-point features. M3C2 is an algorithm that calculates the robust distance between two sets of point clouds directly based on the surface orientation of points without network construction, and the algorithm principle is illustrated in Figure 1. With a reference point as the center of the circle and D as the diameter, the normal vector on the reference point cloud is estimated. The normal vector direction is taken as the cylindrical axis to construct a cylinder with a diameter of d and a height of h. Then, a one-to-one correspondence is established for the two-point clouds. Finally, the point clouds within the cylindrical range are projected to the cylindrical axis. The average position is taken to obtain the M3C2 distance of each point between the two-point clouds. Additionally, the algorithm combines the number of projection point clouds, $n_{i}$, roughness, $\sigma_{i}$, and registration error, reg, between the two-point clouds to define a level of detection (level of detection, $LOD_{x\%}$) for each measurement. The reliability of the distance calculation results is evaluated from the perspective of statistical significance. The reliability evaluation calculation method at the 95% confidence level is expressed in Equation (1).
(1)$$LOD_{95\%}(d)=\pm 1.96\left(\sqrt{\frac{\sigma_{1}^{2}}{n_{1}}+\frac{\sigma_{2}^{2}}{n_{2}}}+reg\right)$$

Since the ZEB-HORIZON hand-held 3D Laser Scanner cannot add ground control points in the acquisition process, ground control points are not laid in the research area. Both TLS point cloud and HLS point cloud adopt independent coordinate system. Therefore, before M3C2 distance calculation, coordinate unification is needed. However, the UAS contains the RTK module, whose image acquisition and point cloud generation are based on the WGS84 coordinate system, and the positioning accuracy can reach centimeter level. Therefore, the study takes the UAS point cloud as a reference and registers the laser point cloud to the WGS84 coordinate system through the way of point cloud matching.

#### 2.1.3. TLS Point Cloud Hole Repair Technology

According to the results of feature comparison and analysis of 3D data, TLS point cloud was used as the basic point cloud. Meanwhile, the remaining two-point clouds were adopted to repair the void to achieve the purpose of improving the integrity and quality [16] of the point cloud. In this study, the roughness and normal vector change rate of HLS point cloud and TLS point cloud were calculated with the same neighborhood radius. The nearest neighbor between the two data was retrieved with the three-dimensional coordinate value. The nearest neighbor point of HLS significant change point extracted in the M3C2 distance calculation was obtained in the TLS point cloud and its geometric eigenvalues. In addition, the difference between the geometric eigenvalues was revealed. The distribution difference was analyzed in 2D, and the main distribution range of the feature difference was extracted with the difference distribution range as a reference. Using the difference between the geometric eigenvalues of all points of the HLS point cloud and the TLS point cloud, more feature points are screened from the HLS point cloud to supplement the number of TLS point cloud feature points. The technical route is illuminated in Figure 2.

The data holes in the TLS point cloud were divided into top voids and facade voids. Concerning the top surface hole, the study projected and rasterized the TLS point cloud from the perspective of a top-down view, taking the point cloud elevation value contained in the cell as the cell value, outputting the missing area of the top surface point cloud with zero cell value accordingly, and further employing the data segmentation method based on Euclidelist clustering [17] and alpha shape [18]. Other algorithms were utilized to screen the top surface point cloud voids and extract the boundary range of the void projection surface. Finally, the hole repair data on the UAS point cloud were extracted using the improved angle method and fused with the TLS point cloud to realize the repair of the holes on the top surface of the TLS point cloud. The technical route is exhibited in Figure 3. Given the façade voids existing in the TLS point cloud, polygon filtering as conducted to extract the façade void areas. With a small number of point clouds in the void area as seed points, the nearest neighbor retrieval was taken to find the k nearest neighbor points of seed points in the HLS point cloud. Additionally, the search results were combined with the data to be repaired to complete the repair of the facade voids.

#### 2.1.4. Overview of the Research Object

In this research, a structure in the Lianhua Campus of Kunming University of Science and Technology in Kunming City, Yunnan Province was selected as the object, with geographical coordinates of about 102.704136° E and 25.067418° N. The structure is located in the center of a rectangular outdoor square with an area of about 187 $m^{2}$, which is an open space and facilitates data collection. The target structure is about 7.41 m high. The base with a diameter of about 3.85 m is composed of a round platform and a prism and is covered with tiles on all four sides. However, there is partial shedding, as well as a small number of weeds on the roof of the base. Above the pedestal is a copper irregular statue. The maximum width of the statue part is about 8.41 m, and the maximum height is 3.75 m. The research object area is presented in Figure 4. The entire structure contains rich geometric and texture information, making it difficult to accurately acquire with traditional data acquisition methods and unable to be expressed in the digital space. Therefore, 3D laser scanning technology and low-altitude photogrammetry technology were employed to acquire the point cloud data of the target structure.

### 2.2. Air–Ground Multi-Source Point Cloud Data Acquisition and Preprocessing

#### 2.2.1. UAS Low-Altitude Photogrammetric Point Cloud Acquisition and Preprocessing

There is no significant obstruction around the research object, and the airspace is open. Hence, a DJI Phantom 4 RTK quadcopter UAV at a relative altitude of 45 m and at a low altitude with 80% heading overlap and 65% side-facing overlap was used as the standard. According to the “井” shaped route, the acquisition of target image data was achieved using a high-spatial resolution optical sensor equipped with the UAV. The route is displayed in Figure 5a.

Image data acquisition of the structure was conducted with the above method. The spatial resolution of the obtained image was about 1.23 cm/pixels. After the image data were encrypted by the Context Capture v4.4 software, the tile range of the structure was selected; the stereoscopic matching of the scene was performed; the depth was obtained; the structure mesh of the target area was constructed; and finally, the mesh was uniformly sampled at 1cm resolution to obtain the UAS point cloud. Subsequently, the ground points, point clouds of outliers, and point clouds of non-structures within the scope of tiles were removed by a CSF (cloth simulation filter) filter [19] and a polygon filter. Around 738,900 UAV photogrammetry point clouds of the target structure were obtained. The different orientation projections of the point cloud are exhibited in Figure 6.

#### 2.2.2. Terrestrial 3D Laser Scanner to Obtain Point Clouds and Preprocessing

In this study, the TLS point cloud was collected from the east, southwest, and northwest directions of the target structure using the MAPTEK I-Site8200ER terrestrial laser scanner in the form of three stations, as rendered in Figure 3b. Each time the station acquired a point cloud data set. After the original data acquisition was finished, the original data were separately polygonal filtered, the points outside the study area and a small amount of noise were filtered, and then the point cloud coarse matching was performed using the same name point pair between different data. Then, exact matching was performed by ICP (iterative closest point) [20]. The registration accuracy of two pairs of data between the three stations was controlled at about 1cm. After processing, a total of about 796,100 TLS point clouds were obtained. The obtained TLS point cloud projection view is provided in Figure 7.

#### 2.2.3. Ground Hand-Held 3D Laser Scanner to Obtain Point Clouds and Preprocessing

The experimenter lifted the HLS scanning head to face the direction of the structure about 5 m from the edge of the base of the structure and performed HLS point cloud data acquisition on the target structure in a walking state according to the scanning path shown in Figure 5c. During the data acquisition, the motor-driven scanning head on the HLS was continuously rotated to obtain all data within the range of pitching angle 270° and tumbling angle 360° centered on the scanning head. Afterwards, non-structured point clouds in the data were rejected by polygon filtering. Then, the average distance estimate was made in groups of five points, and the SOR (Statistical Outlier Filter) [21] was used for filtering by 1 Sigma. Approximately 410,000 outliers in the HLS point cloud were excluded using the average distance of the point set. Duplicated point clouds with a minimum distance of 0.5 mm between point coordinates were removed to reduce data redundancy. Only unique data were retained under the same coordinate value. A total of 3,333,400 points of data were obtained. The projection of the HLS point cloud from different directions is presented in Figure 8.

## 3. Result and Discussion

### 3.1. Comparative Analysis of Multi-Source Point Cloud Features

#### 3.1.1. Comparative Analysis of HLS and TLS Point Cloud Features

HLS and TLS were adopted to collect the laser point cloud of the target. Although both instruments use a near-infrared laser as the scanning light source, the acquisition method and the point cloud 3D coordinate calculation method are completely different. Moreover, there are significant differences in the point cloud obtained from the same structure. The average density of the HLS point cloud is 0.932 $points/cm^{3}$, which is much larger than the 0.356 $points/cm^{3}$ of the TLS point cloud [22]. Therefore, the HLS point cloud has a richer amount of data. Due to the existence the occlusion of the structure itself, the obtained TLS point cloud still lacked data in multiple areas and angles, though the TLS point cloud data acquisition scheme was designed according to the shape of the structure before the study. With the purpose of ensuring that the TLS point cloud did not have any data missing within its field of view, it was necessary to repeatedly add multiple stations to collect point clouds and match the multi-site cloud data in real-time. This was to complete the combination of the cloud of each measurement site of the structure for achieving the integrity of the point cloud. This considerably increases the number of field operations and reduces the efficiency of operations. The increasing number of stations indicates an increase in the number of matchings. Meanwhile, the overall accuracy of the point cloud decreases, and the increase in stations leads to a repeated collection of point cloud data in the same area, resulting in the redundancy of point cloud data. HLS was used to complete the collection of point cloud data in the move and flexibly obtain data from multiple angles. Since the scanning process does not require strict path planning, the position of the scanning head can be actively adjusted in the data collection process. Furthermore, the special-shaped structure is prone to data vulnerability areas for key scanning, the integrity of the point cloud can be more guaranteed, and the obtained data can be adopted to check whether there is data missing in the point cloud without any processing. Even though there is a point cloud missing, the location of the missing data can be determined. A new scanning scheme was quickly developed, and data collection was re-performed, so as to complete the missing target point cloud more quickly than TLS. The complete pair of the two-point clouds is exhibited in the area within the red frame in Figure 9.

However, HLS mainly relies on lidar, IMU, and an accelerometer to obtain incremental point cloud data. The cumulative error of three-dimensional coordinates can be reduced by loop detection, while the existence of cumulative error will still reduce the coordinate accuracy of the HLS point cloud. Moreover, the TLS point cloud three-dimensional coordinate calculation method does not have such cumulative error. Thus, the point cloud obtained by TLS has higher precision position information. Additionally, the roughness of the two data sets was calculated by the same adjacent radius, and the roughness of the HLS point cloud data was close to twice that of the TLS point cloud, as illustrated in Figure 10. Owing to the data drift present in the HLS point cloud data, the roughness of the same surface HLS point cloud far exceeds that of the TLS point cloud. In subsequent point cloud applications such as model construction and contour extraction, it is not possible to accurately locate the texture feature points and boundary feature points of the structure in the HLS point cloud, resulting in distortion of the final result. Therefore, the TLS point cloud was adopted as the basic data for these two laser point clouds. The HLS point cloud was screened according to the data missing range and texture information differences in the TLS point cloud, and the filtered data was merged to achieve the effect of improving the TLS point cloud data.

#### 3.1.2. Comparative Analysis of UAS Point Cloud and TLS Point Cloud Features

The method of low-altitude photogrammetry was employed to obtain the UAS photogrammetric point cloud of the structure from the air. The UAS point cloud is noticeably different from the TLS point cloud in that the photogrammetry point cloud data contains color information. However, it cannot represent small components on the structure (Figure 11a). Moreover, it cannot represent the grooved texture on the side of the structure, and the wrong point cloud data is generated at the hole of the structure (Figure 11b). In Figure 11c, the red box indicates the weed-covered part of the structure, and the optical sensor used in the UAS photogrammetric point cloud cannot penetrate the vegetation. TLS is not affected by this; the emitted laser can penetrate the vegetation and directly obtain the point cloud on the surface of the structure. Considering that UAS photogrammetric point clouds do not provide a good representation of side structure information, elevation point clouds in photogrammetry point clouds can be eliminated when different types of point cloud data are merged.

However, the top view of the different point clouds in Figure 6a, Figure 7a, and Figure 8a reveals that only the UAS photogrammetric point cloud contains the top surface information of the structure more completely, and both laser point clouds have large point cloud holes in this part, which can only demonstrate the top surface boundary information. Therefore, photogrammetric point clouds can be used to patch up holes in this section.

#### 3.1.3. Comparative Analysis of Point Cloud Features Based on M3C2 Distance

With the TLS point cloud as a reference, the above comparative analysis between different point clouds describes the differences between different point clouds from the perspective of local geometric characteristics. Taking TLS as the reference point cloud, the M3C2 distance of three-point cloud data was calculated from the perspective of single-point geometric features to further unveil the characteristic differences between different point clouds. Before the calculation, the methods of point-to-coarse matching and ICP exact matching were adopted to unify the spatial position of the point clouds acquired by different sensors. In addition, the registration mean squared error between different data was controlled within 3cm to avoid the impact of the registration error on the M3C2 distance calculation results. Meanwhile, the registration error was introduced when the M3C2 distance reliability assessment is conducted. Notably, the data registration was not scaled to any point cloud for retaining the original characteristics of different data.

In the M3C2 distance calculation, the normal vector was estimated with the method of multi-scale estimation to ensure that the normal vector of the tiny surfaces in different point clouds was not smoothed, to weaken the influence of the point cloud roughness on the direction of the normal vector and to guarantee the correctness of the direction when looking for the comparison point cloud. Specifically, the opposite side of the centroid was utilized to define the normal vector orientation, and the estimated radius of the normal vector was changed in steps from 0.05m to 0.55m. Finally, the normal vector estimation was completed. After the normal vector estimation, the M3C2 distance was obtained with 0.1m as the cylindrical height and 0.04m as the cylindrical radius. A total of two sets of calculation results were obtained through the calculation of M3C2 distance between the two-point clouds. Table 3 provides the basic statistical parameters of M3C2 distance calculated between different data. The average distance can be observed to be millimeters between the photogrammetric point clouds and the TLS point clouds. The average M3C2 distance between the HLS point cloud and TLS point clouds is larger than 1 cm. Additionally, the M3C2 distance uncertainty was fitted following the Gaussian distribution. It was uncovered that the reliability estimation means of all points of M3C2 distance in the two groups are 0.072 and 0.062, and the standard deviations are 0.0048 and 0.0034, respectively. The reliability estimation means and standard deviations of M3C2 between TLS and UAV are relatively low.

The M3C2 distance algorithm also defines the number of projected points, $n_{i}>$4, when calculating the average point. The M3C2 distance at the same point is defined as a significant change point outside the value range of $LOD_{95\%}$, which is a set of points with significant changes on two sets of point clouds. The M3C2 distance calculation results for these point sets have high reliability. According to the statistics, the absolute value of the M3C2 distance of the significant change point is higher than 0.06 m, verifying that there are different expression effects of different point clouds for some areas of the structure. Regarding a more intuitive analysis, the study superimposed and filtered out significant change points on a UAS photogrammetric point cloud with color information to present the data from three angles, as illuminated in Figure 12. Compared with the TLS point clouds, the photogrammetric point clouds have a large number of significant change points in the side-banded areas with complex textures and vegetation occlusion areas. This further suggests that the UAS photogrammetric point clouds have a distortion in the expression of the information of the side of the structure. Moreover, the wrong point cloud is generated, ascribed to the inability to penetrate the vegetation on the building. On the underside of the tail of the structure, it is not possible to fully obtain the photogrammetric point cloud owing to the limitation of the field of view of the optical sensor, contributing to some significant change points. However, most of the photogrammetric top point clouds are not involved in the calculation of M3C2 distance because of the large amount of information missing at the top of TLS point cloud data. The statistical results of significant change points (Figure 13) demonstrate that most of them are concentrated in the sculptural parts of structures with more complex grooved textures, though the number of significant change points between TLS point clouds and HLS point clouds is significantly reduced. In the face of complex trench textures, there are also certain differences between the point clouds collected by TLS and HLS; HLS, with its rich point cloud acquisition perspective and 2.6 times higher point cloud density than TLS point clouds, may contain some feature cloud data that TLS has not collected.

### 3.2. TLS Point Cloud Data Hole Repair Technology

#### 3.2.1. HLS Point Cloud Feature Point Selection

The quality of the HLS point cloud is lower than that of TLS point cloud. However, the analysis of significant change points extracted in the M3C2 distance algorithm demonstrates feature points that cannot be expressed in some TLS point clouds in contrast to HLS point clouds. Hence, from the extraction of HLS point cloud data to deduce significant change points and to fuse into TLS point clouds, more structure feature points can be expressed to enhance the quality of TLS point clouds. However, some missing feature points exist, though the M3C2 distance algorithm can extract some significant change points. The distance is calculated based on the average position of the point set projection. As a result, the algorithm cannot regard these points as significant change points when there are only a few feature points in the area. As illustrated in Figure 14, the M3C2 distance calculated by the point cloud in (a) is equal to that of the point cloud in (b) with the same normal vector estimating the radius, projected cylindrical radius, and the number of projected points. According to Equation (1), the $LOD_{95\%}$ of point cloud (a) has a smaller range of values. Under the determination conditions of the significant change point, the point cloud (a) is more likely to be regarded as a significant change point. Since the comparison point cloud (b) has a higher roughness, it is not easily regarded as a significant change point. Under the determination condition of the significant change point, in some areas with low point density, it is impossible to meet the condition that the number of projection points required by the significant change point, $n_{i},>$ 4. Consequently, some feature points may be ignored in the M3C2 algorithm.

The 3005 significant change points extracted in the above study were taken as samples to further extract more feature points in the HLS point cloud. As observed from the location distribution of the significant change points, the significant change points are distributed in the areas with high normal vector change rate eigenvalues and roughness eigenvalues regardless of the kind of point cloud. These areas contain the key information of the structure texture. Thus, these two geometric eigenvalues are used as the key information for subsequent feature point extraction. However, if the geometric feature value of the significant change point on the HLS point cloud is only adopted as the filter condition, the filtered point cloud may have similar or identical geometric characteristics with the adjacent points on the TLS point cloud; if this part of the point cloud is integrated with the TLS point cloud, it will not only improve the data quality of the TLS point cloud but also cause data redundancy.

Therefore, the roughness and normal vector change rate of HLS point cloud and TLS point cloud were calculated with 5cm as the unified neighborhood radius. A K-D tree was established according to the coordinates of the significant change point, and the nearest neighbor search was performed based on FLANN [23]. The nearest neighbor value of the significant change point in different data and its two geometric feature values were obtained. The results demonstrated that the geometric feature differences between adjacent points were further calculated with the roughness difference as the abscissa coordinate and the normal vector change rate difference as the ordinate coordinate. Additionally, the distribution of the characteristic difference of each significant change point on different point clouds was plotted, and is shown in Figure 15. A 2D kernel density analysis [24] was performed on the scatter plot to further explore the distribution law of the difference between the eigenvalues of significant change points. By extracting the distribution boundary of the different eigenvalue of points with significant changes and using the improved Angle method [25], the geometric feature difference data of all adjacent points of the TLS point cloud and the HLS point cloud was screened out. Finally, the corresponding feature cloud in the HLS point cloud was extracted based on the difference data.

In this study, 65,496 points were extracted, including significant change points. The extracted points were mostly located in the sculpture area of the upper part of the structure. This part of the area has rich trench texture features. In the experiment, CloudCompare V2.6 software was used to merge the extracted feature point cloud and TLS point cloud, and the fusion effect is shown in Figure 16. By comparing the point clouds obtained by different fusion methods, it can be seen that direct fusion of two kinds of point clouds will lead to a large number of point clouds in the smooth surface area, which will cause data redundancy.

The boundary feature points of the three point clouds were extracted using the boundary estimation based on the normal vector [26] to further verify the variation of the feature points in the fusion data. In order to obtain a better extraction effect, the point cloud resolution calculation was completed before boundary estimation. According to the resolution, the normal vector estimation radius and search radius of the TLS point cloud and point cloud merged by feature points were set to 0.0745 m. The normal vector estimation radius and search radius of the direct merging of point clouds was set to 0.0547 m. The value is about 10 times the mean resolution of the point clouds. Under such conditions, better boundary extraction results can be obtained, and the boundary feature point extraction results are illuminated in Figure 17. A total of 87,453 boundary feature points were extracted from the TLS point cloud, and this is 3491 points less than the boundary feature points extracted from the merged point cloud. Moreover, the number of point cloud boundary feature points after fusion is improved. Although direct merging of point clouds can extract more boundary feature points due to the addition of all HLS point clouds, the boundary feature extraction effect in the small component area at the top and the vegetation covered area in the middle is even worse than the TLS point cloud.

#### 3.2.2. TLS Point Cloud Top Surface Hole Repair

Through the merging of TLS point clouds by the above methods, the side grooved texture and boundary characteristics of the structure can be expressed in detail. Nevertheless, a large number of data are missing on the top surface of the structure. Although the UAV photogrammetry point cloud data has a poor effect on the expression of lateral texture information, it can better express the missing top surface information in the TLS point cloud [27]. As observed from the top view of the TLS point cloud, the current data loss is mainly concentrated in the top surface of the prism and the top surface of the sculpture. Without considering the z-value of the point cloud, the data loss range of the top surface of the sculpture coincides with the range of the data loss of the top surface of the frustum. In addition, in the overhead state, the point clouds at both ends of the sculpture will cause partial obstruction of the hole on the top surface of the frustum, forming an illusion of no data loss, as shown in Figure 18.

Therefore, the point cloud was first sliced along the junction of the base and the statue. Then, part of the point cloud of the sculpture was removed. Afterwards, the data occlusion was eliminated without changing the range of the point cloud cavity. Next, being reversed along the Z axis, the TLS point cloud data was projected by point cloud rasterization. The raster resolution was 0.02m. The raster cell value was the maximum height value of the included point cloud. If there was no point cloud in the cell, the cell value was zero. The point cloud rasterization result is presented in Figure 19a. After the rasterization of the point cloud data, the pixel center grid was output as a point cloud, with the height value information (Figure 19b). Finally, the height value information was employed as a filtering condition to extract the center point data of the hole area raster. The zero-valued cell center points obtained in the study were some discrete sets of points. Nonetheless, some areas with low point density were misjudged as point cloud data holes due to the high raster resolution. Meanwhile, data blank areas outside the structure were extracted with the method. Therefore, the real cavity was divided into non-continuous cavity regions to distinguish the misjudgment region from the real cavity. Moreover, basic data conditions were provided for the extraction of the cavity boundary. The Euclidean clustering data were divided with 4cm as the search radius and 25 as the minimum number of points required for clustering. The segmentation results are demonstrated in Figure 19c. Through the setting of the above thresholds, the low-density false positive void was removed in the segmentation results, and the hole was divided into 14 independent areas, including four data blank areas other than the structure. In addition, the four blank area data were deleted according to the partition result index, and the point cloud data hole area in the top view was obtained.

At this time, the hole area data obtained z value was 0, only containing x and y value information. The above alpha shape and other algorithms were employed to extract the boundary range of x and y values. Any hole boundary can be obtained. In the case of no consideration of z, according to the hole boundary, the improved angle method was adopted to filter the data of UAS photogrammetric point cloud x and y values. Then, part of the UAS photogrammetric point cloud was obtained, including the point cloud required for hole repair. However, there were also some low-quality elevation data in this point cloud because the z-value during data filtering was ignored. Given this complication, the point cloud of the diagonal surface was further filtered per the elevation value. The final obtained point cloud could make up for the lack of most of the top surface data, as illustrated in Figure 20b.

#### 3.2.3. TLS Point Cloud Facade Hole Repair

It is necessary to set up multiple point clouds at multiple fixed sites with different fields of view when TLS collects the point cloud of a structure. The obtained point clouds need to be spatially matched to reflect the 3D information of the structure [28,29,30]. Especially in the face of the goal of complex structure, TLS must ensure the integrity of point cloud acquisition by increasing the number of stations. Nonetheless, an increase in the number of stations not only lowers the operation efficiency and overall accuracy of the data, but also causes a large number of point cloud data redundancies [31]. If the station layout is unreasonable, there will be data holes after the point cloud matching, as exhibited in the red frame range in Figure 9a. It is difficult to detect these small-area voids using TLS for field operations. Moreover, data recollection of these small-area voids significantly boosts the cost of operation. HLS collects the point cloud data of the structure in motion and can continuously adjust the direction of the scanning head during the acquisition process to collect data on all areas of the structure. There will be almost no point cloud holes as long as it is within the reach of the scanner’s field of view. Additionally, HLS point clouds can obtain three-dimensional information of the structure without point cloud matching, they can check the integrity of point cloud data on-site, and they can quickly develop new scanning methods to re-collect point cloud data and obtain complete target point cloud data if there is a point cloud hole. Therefore, HLS point cloud data was used to repair the small areas of façade holes in the TLS point cloud data.

Although the extracted HLS feature points were fused with the TLS point cloud in the above study, the HLS feature points could not achieve the effect of repairing the cavities in the TLS point cloud façade. Hence, this study first adopted polygon filtering to extract the cavities of the façade on the fused point cloud. Nevertheless, more boundary points appeared in the extracted data (Figure 21a). Thus, the point cloud density was calculated with 5cm as the field radius of the part of the data, and the part of the high-density boundary point was excluded. The results are presented in Figure 21b. The remaining low-density point cloud was taken as the seed point. The FLANN-based nearest neighbor search was performed in the HLS point cloud. In addition, k nearest neighbors of the seed point in the HLS point cloud were observed. The retrieval results were fused with the data to be repaired, and the repair of the facade void could be completed, as demonstrated in Figure 21c.

In the fast nearest neighbor search based on FLANN, the approximate points that can be extracted increase as the k-value increases; the number of near-neighbors increases with the product of the number of seed points and the k-value increment. In this study, 25 steps were taken as the length. In the range of 25–150 intervals, the k value was adjusted, and the neighbor point search was performed separately and calculated the point cloud density with 3 cm radius, as shown in Figure 22.

By comparing the point cloud density retrieved by different k-values with the mean value of HLS point cloud density, it can be revealed that different elevations and voids should use different k-values for HLS point cloud retrieval. Only in this way can we ensure that the retrieved point cloud density is close to the HLS point cloud density, and no data beyond the hole will be retrieved due to excessive retrieval. For holes a and b, only when k = 100 was the retrieved point cloud density higher than that of the HLS point cloud density, at which point 5800 points and 24,000 points were extracted, respectively. When k = 25, 5250 points and 20,200 points could be retrieved from holes c and d. At this time, the point cloud density has met the demand for hole repair. The effect picture of repairing the facade hole is rendered in Figure 23, where the blue point cloud indicates the point cloud extracted from HLS to repair the facade hole.

## 4. Conclusions

In this study, TLS point cloud, HLS point cloud, and UAS point cloud data of a special-shaped structure on the studied campus were acquired in three ways. In order to achieve the purpose of difference analysis and cavity repair of multi-source point clouds of the same structure, the geometric characteristics of point cloud data were compared and analyzed from single-point aspects. According to the results of comparative analysis, a point cloud was selected as the reference point cloud. The method of multi-source feature difference analysis, hole location, and neighborhood point retrieval was used to screen the relevant point clouds of the hole part of the reference point cloud from the other two kinds of point clouds, and then fused with the reference point cloud to improve the quality of the reference point cloud. According to the comparative analysis results, the TLS point cloud with high accuracy, low roughness, and small systematic error was selected as the reference point cloud. Based on the characteristic difference between the significant change points extracted from the M3C2 distance on different point clouds, 65496 feature points were screened out from the HLS point cloud. The geometric feature information contained in the TLS point cloud was enriched through integration with the TLS point cloud. The optimized TLS point cloud can express the characteristics of the structure boundary and texture in more detail. The noticeable data holes in the TLS point cloud were distinguished into two types: top surface and facade for repair. Additionally, the repair of the TLS point cloud top surface hole was realized using UAS photogrammetric point clouds, and a good top surface hole repair effect was achieved by means of a point cloud to the grid. For the facade cavity area, a very small number of point clouds existing in the TLS point cloud hole area were used as the seed point. Neighborhood retrieval was performed on the HLS point cloud. The search results were employed to repair the facade hole. The number of point clouds in the two facade hole areas was increased by about 28 times and 100 times, respectively.

The structure point cloud data obtained in different ways all contain the geometric characteristics of the structure. Nonetheless, there must be some information differences between the different point clouds. Through the comprehensive utilization of three-point clouds, the TLS point cloud hole was repaired, and its geometric characteristics were strengthened, achieving the expected purpose of our study. However, the method used in this paper did not make full use of the point cloud data elevation value to eliminate invalid elevation point clouds in the range when extracting point cloud data within the top surface cavity range. In addition, the patching point cloud density distribution in the facade hole area was uneven. Therefore, higher dimensional data will be utilized to accurately determine the range of holes and reasonably extract the data in the range from other point clouds in future research.

## Figures and Tables

**Figure 1 sensors-22-09627-f001:**
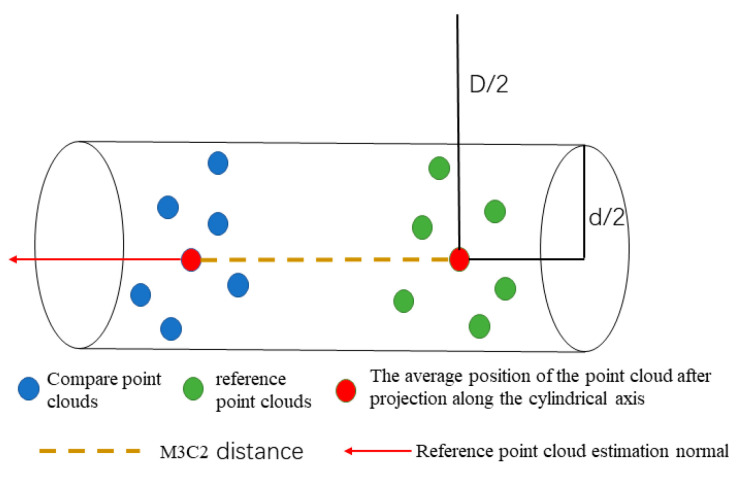
Schematic diagram of M3C2 distance algorithm.

**Figure 2 sensors-22-09627-f002:**
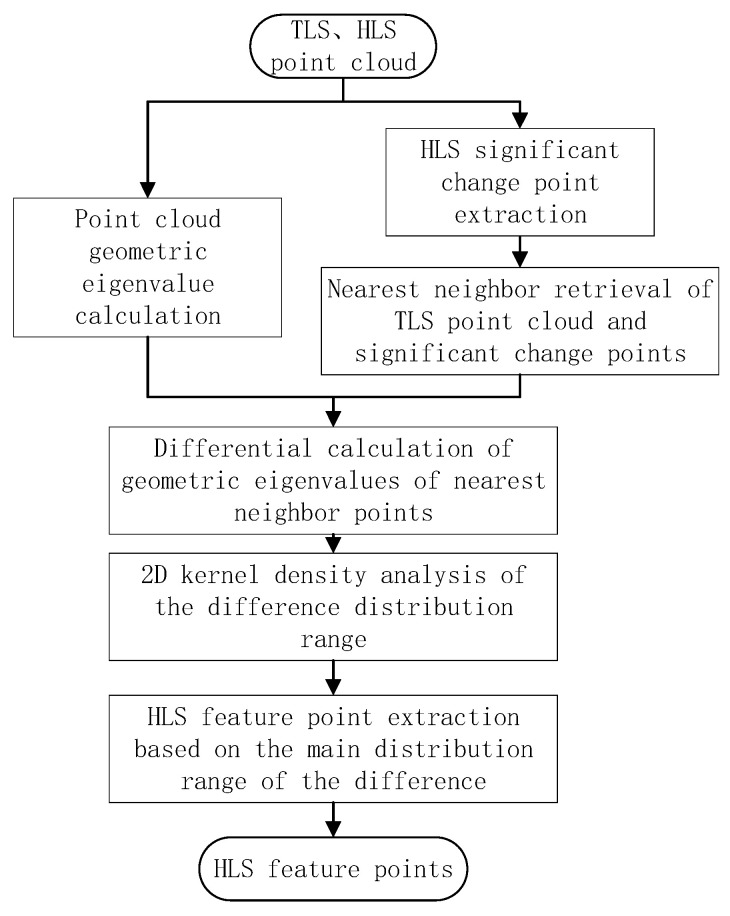
HLS point cloud feature point filtering flowchart.

**Figure 3 sensors-22-09627-f003:**
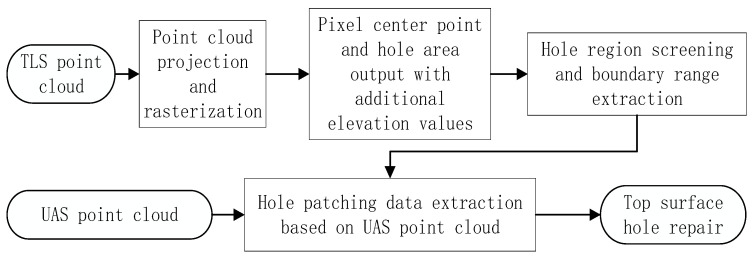
TLS point cloud top surface hole repair flowchart.

**Figure 4 sensors-22-09627-f004:**
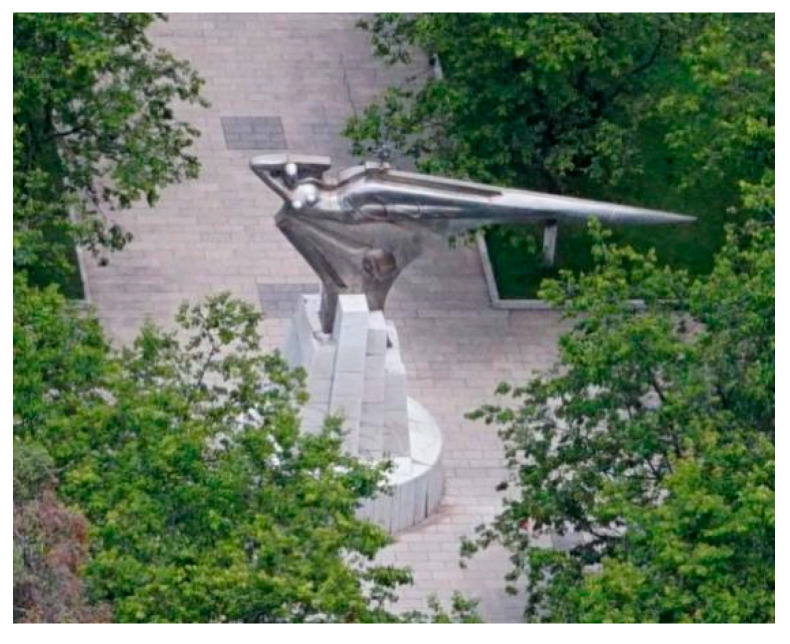
Status of the study area.

**Figure 5 sensors-22-09627-f005:**
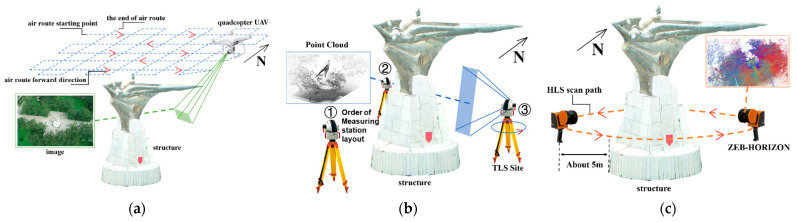
Schematic diagram of each data acquisition path. (**a**) Schematic diagram of the aerial survey route. (**b**) Schematic of the TLS scanning site. (**c**) Schematic of the HLS scan path.

**Figure 6 sensors-22-09627-f006:**
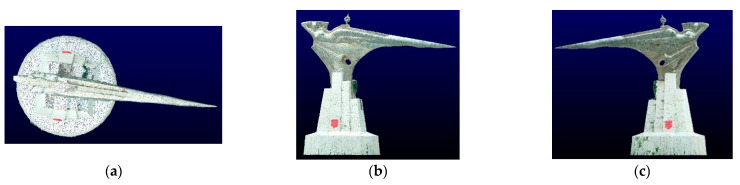
UAS photogrammetric point cloud projection view obtained by the laboratory. (**a**) Top view. (**b**) Front view. (**c**) Rear view.

**Figure 7 sensors-22-09627-f007:**
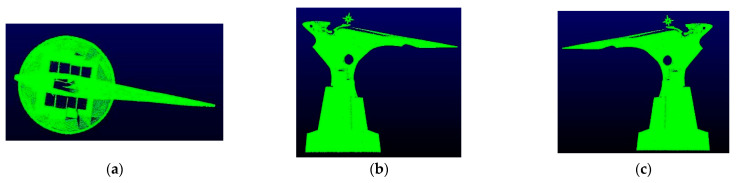
Projected view of TLS point cloud in different directions. (**a**) Top view. (**b**) Front view. (**c**) Rear view.

**Figure 8 sensors-22-09627-f008:**
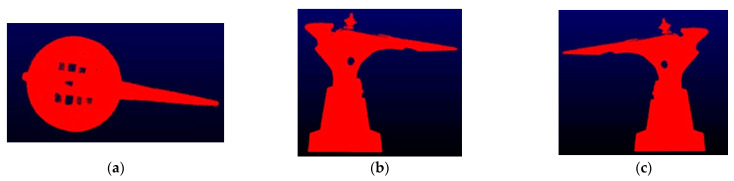
HLS point cloud projection view obtained in different directions. (**a**) Top view. (**b**) Front view. (**c**) Rear view.

**Figure 9 sensors-22-09627-f009:**
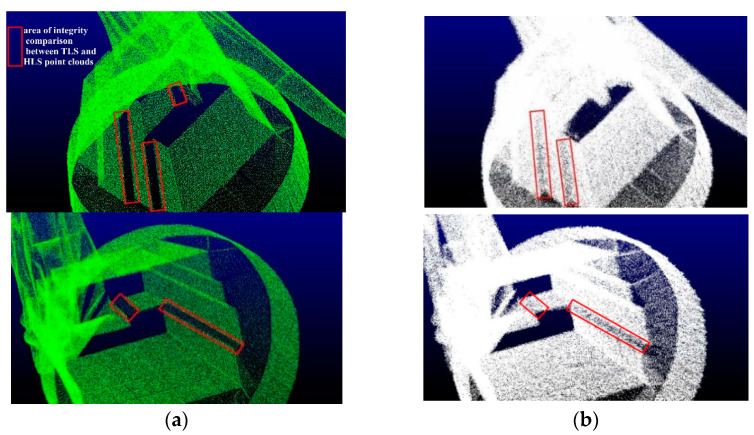
Data integrity comparison between TLS and HLS point clouds. (**a**) TLS partial point cloud. (**b**) HLS partial point cloud.

**Figure 10 sensors-22-09627-f010:**
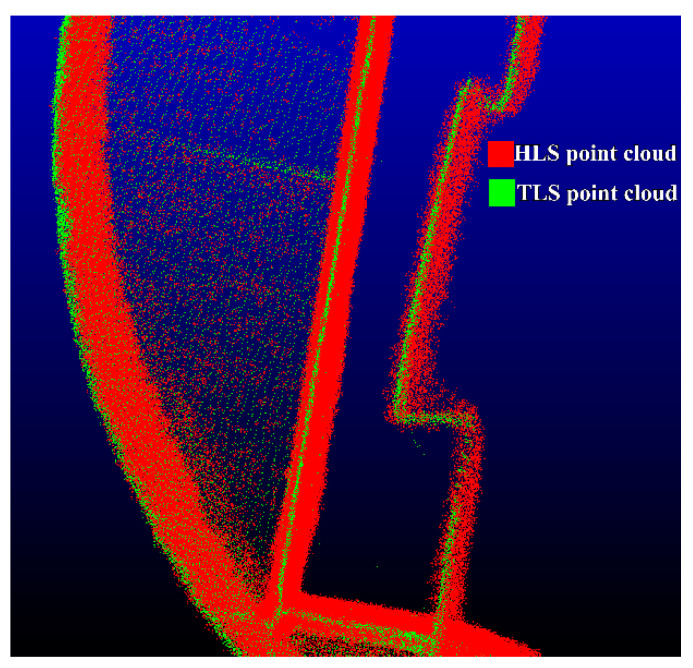
Cross-sectional view of HLS and TLS point cloud data.

**Figure 11 sensors-22-09627-f011:**
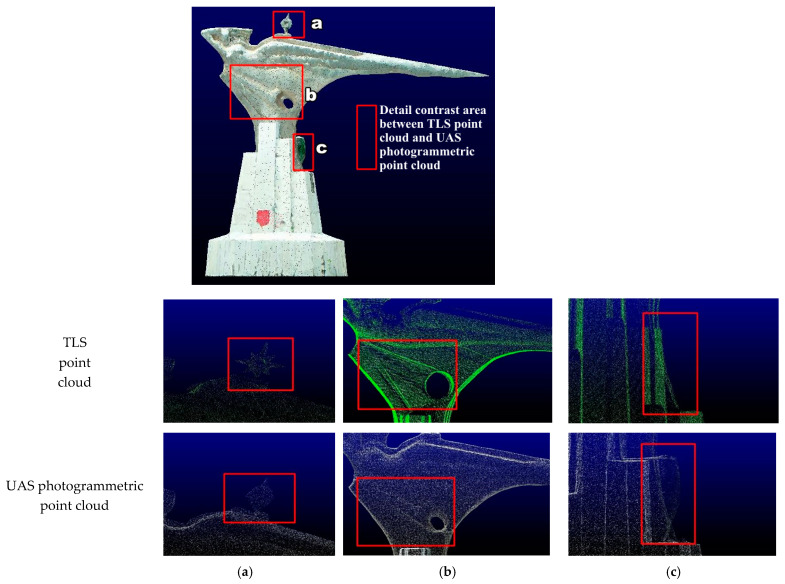
Comparison of TLS point cloud and UAS photogrammetric point cloud details. (**a**) Small component point cloud. (**b**) Grooved texture point cloud. (**c**) Point cloud in vegetation-covered areas.

**Figure 12 sensors-22-09627-f012:**
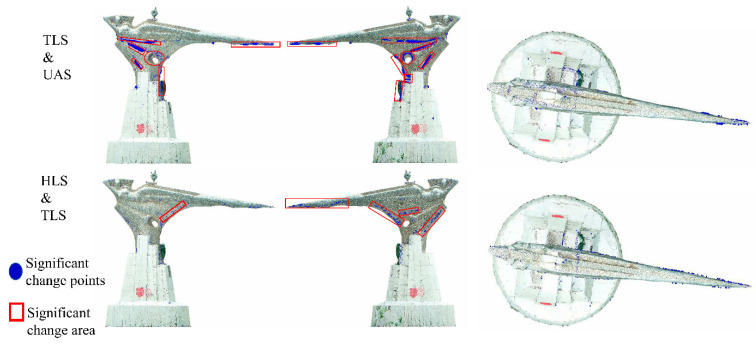
Distribution of significant change points.

**Figure 13 sensors-22-09627-f013:**
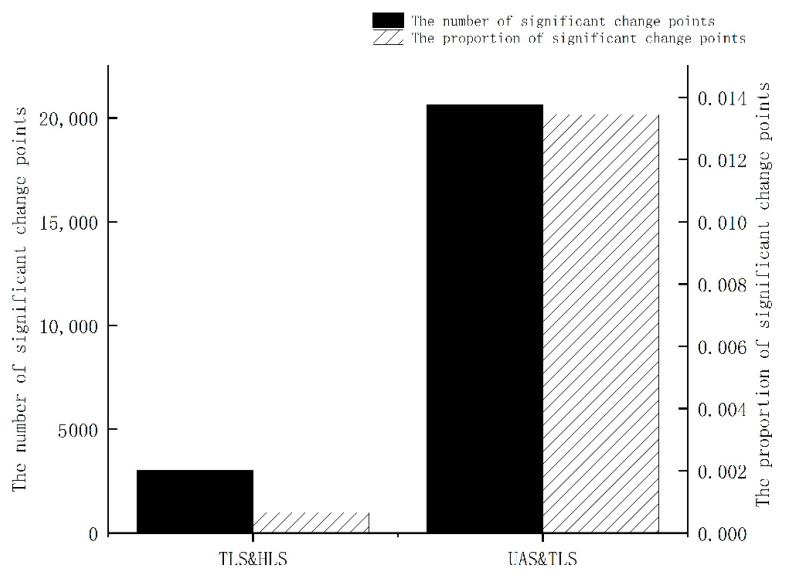
Statistics on the number and proportion of significant change points.

**Figure 14 sensors-22-09627-f014:**
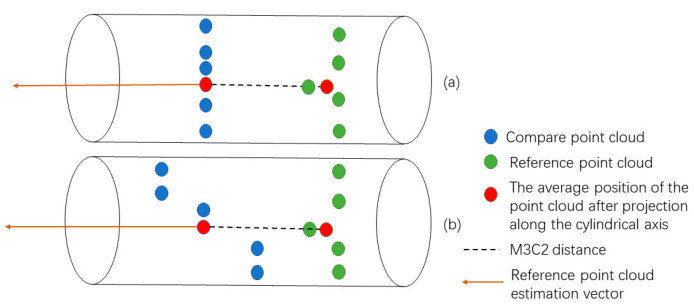
Comparison of M3C2 distance calculation effects of different point clouds. (**a**) M3C2 distance calculation results of low roughness compare point cloud. (**b**) M3C2 distance calculation results of high roughness compare point cloud.

**Figure 15 sensors-22-09627-f015:**
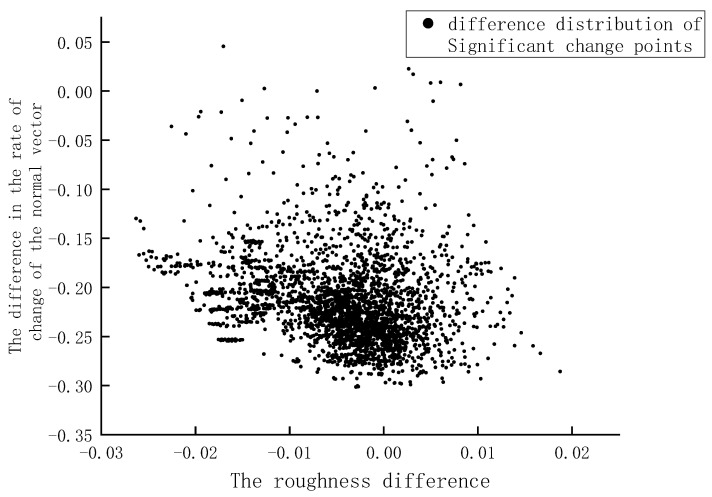
Significant change point characteristic difference distribution.

**Figure 16 sensors-22-09627-f016:**
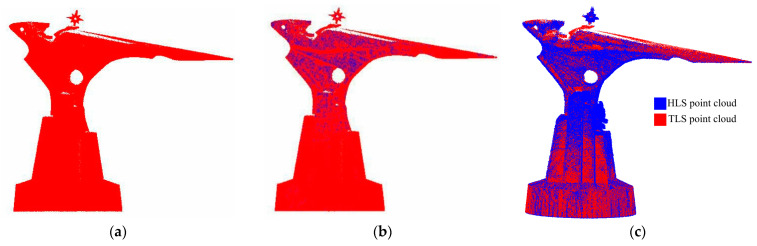
Rendering of point cloud fusion. (**a**) TLS point cloud. (**b**) Point cloud merged by feature points. (**c**) Direct merging of point clouds.

**Figure 17 sensors-22-09627-f017:**
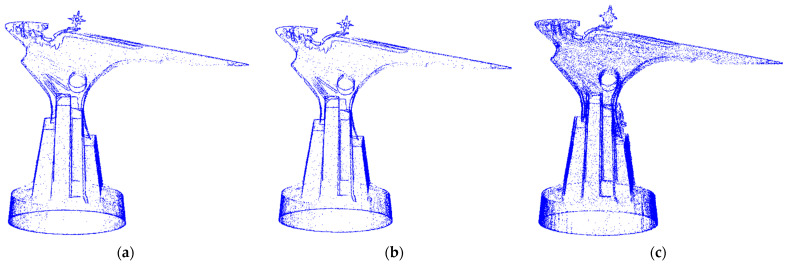
Boundary feature point extraction results. (**a**) TLS point cloud boundary extraction results. (**b**) The boundary extraction results of point cloud fusion extracted by the research method. (**c**) Result of boundary extraction after direct fusion of point clouds.

**Figure 18 sensors-22-09627-f018:**
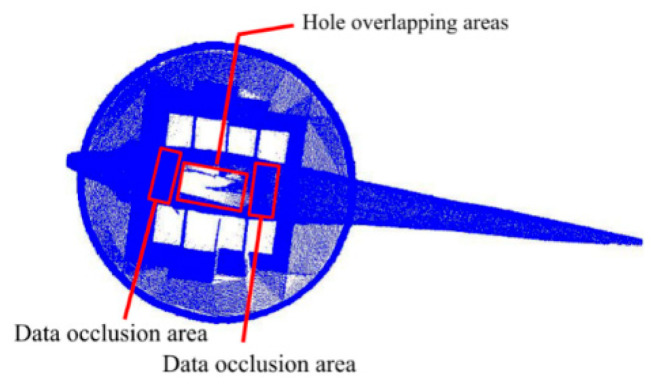
Schematic of data occlusion and hole overlapping areas.

**Figure 19 sensors-22-09627-f019:**
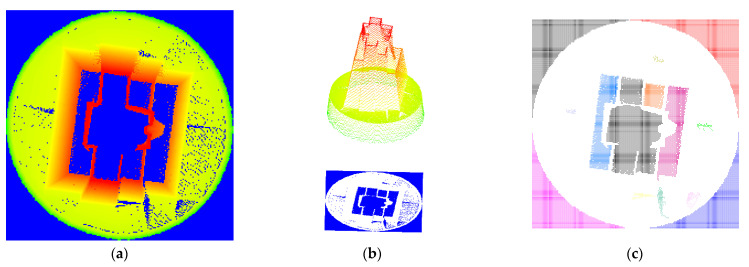
TLS point cloud hole data acquisition. (**a**) Point cloud rasterization. (**b**) A cell center point cloud with an attached elevation value. (**c**) The result of the cell center point cloud splitting.

**Figure 20 sensors-22-09627-f020:**
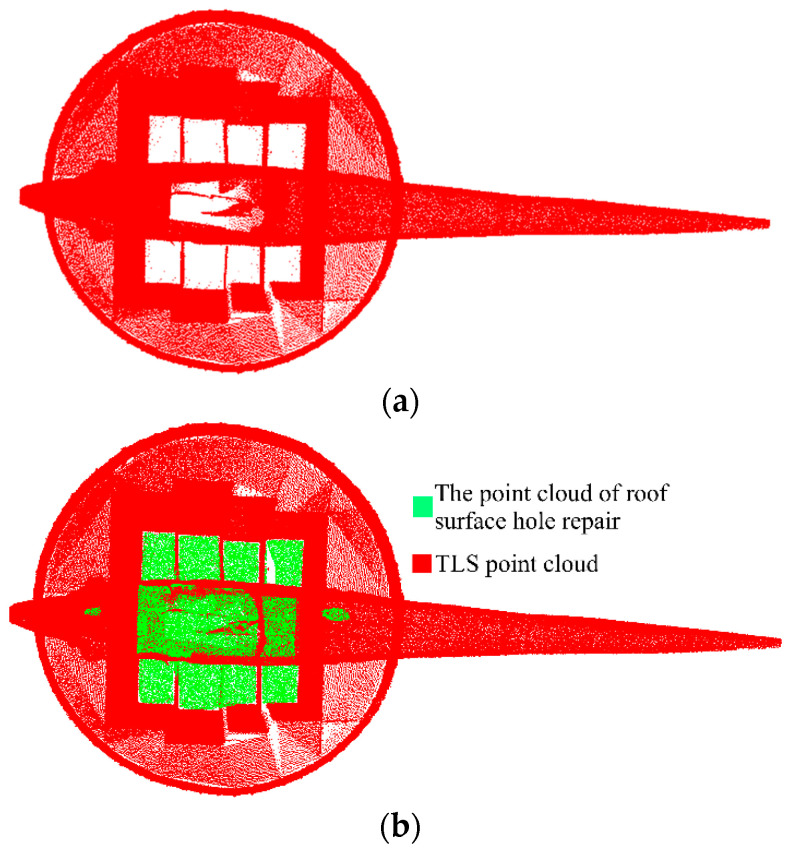
TLS point cloud roof surface hole repair. (**a**) TLS point cloud top view. (**b**) Top view of the TLS point cloud patched by the hole in the top surface.

**Figure 21 sensors-22-09627-f021:**
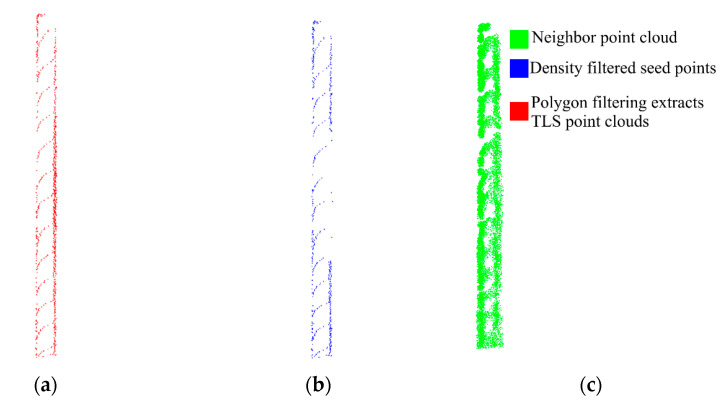
Facade hole data extraction. (**a**) Polygon filtering extracts point clouds. (**b**) Density filtered seed points. (**c**) Neighbor search results.

**Figure 22 sensors-22-09627-f022:**
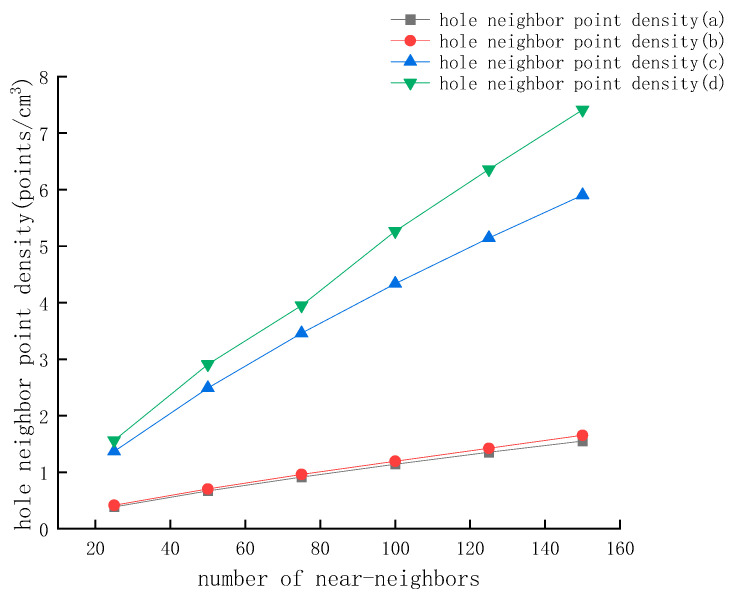
Point cloud density retrieved with different K values. (**a**) Point cloud density retrieved in hole a with different K values. (**b**) Point cloud density retrieved in hole b with different K values. (**c**) Point cloud density retrieved in hole c with different K values. (**d**) Point cloud density retrieved in hole d with different K values.

**Figure 23 sensors-22-09627-f023:**
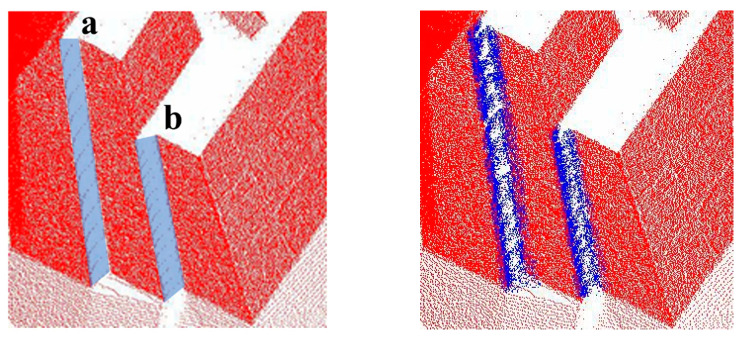
TLS point cloud facade patching effect. (**a**) The facade hole of the TLS point cloud. (**b**) TLS point cloud facade hole repair effect. Red point cloud is TLS point cloud, the blue point cloud indicates the point cloud extracted from HLS to repair the facade hole. The ‘a b c d’ in the figure is the hole number in reference to Figure 22.

**Table 1 sensors-22-09627-t001:** Parameters of the 3D laser scanners.

	Instrument Appearance	Maximum Range	Maximum Scanning Speed	Weight	The Field of Scanning View
MAPTEK I-Site8200ER	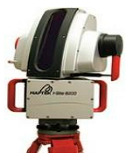	750 m	80,000 points/s	15 kg	360 × 250
GeoSLAM ZEB-HORIZON	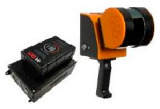	100 m	300,000 points/s	2.76 kg	360 × 270

**Table 2 sensors-22-09627-t002:** Optical sensor parameters.

Instrument Appearance	Sensor Size	Valid Pixels	ISO Range	The Maximum Resolution of the Photo
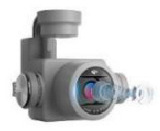	1 inch	20 million	100–6400	4864 × 3648 (4:3) 5472 × 3648 (3:2)

**Table 3 sensors-22-09627-t003:** Basic statistical parameters of M3C2 distance based on TLS point cloud as reference data (m).

		Average Value	Maximum Value	Minimum Value	Standard Deviation
Compared data	HLS	−0.014399	0.107545	−0.094519	0.0180320
	UAV	0.001059	0.139461	−0.1658	0.0184823

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
