# Peer review of "Feature Analysis of Scanning Point Cloud of Structure and Research on Hole Repair Technology Considering Space-Ground Multi-Source 3D Data Acquisition"

_sensors, 2022, doi:10.3390/s22249627_

Round 1

Reviewer 1 Report

The authors proposed a method for combining the point clouds acquired by three different technologies. While the topic is interesting, I have some major concerns regrading the manuscript. First of all, the manuscript is poorly organized and there is not a consistent path to follow the direction. Second, there are two proposed flowchart in the manuscript without sufficient explanation, while there are a lot of junk information. I also have a concern about the motivation. By increasing the number of TLS stations, more complete and accurate point cloud may be obtained. Do we need HLS in this case at all?

The rest of my concerns are listed as follows:

1- The abstract is not informative, it should be rewritten.

2-the structure of the paper should be mentioned at the end of introduction section.

3-The authors stated that: "the spatial resolution of the obtained image is about 1.23cm/pixels.", line 214. However, in line 216 mentioned that "a point cloud is generated at a resolution of 1cm". Is any interpolation applied to raw data?

4-Line 216: what the authors mean by "obvious noise"?

5-For the HLS point cloud, the authors mentioned that the point cloud is obtained by walking around the object. I am wondering how the top-view is generated in this case.

6- line 239: Please explain this: " by removing duplicate points from the data under the condition that the minimum distance between the two points is 0cm"

7- line 249: "average density of the HLS point cloud is 105.45 ??????". Usually the point density is stated in "points per area" or "points per volume".

8-line 249: Does having different point densities imply that the local geometry of two point clouds are different?

9-Figure 7: Acquiring more texture in HLS is not about the nature of measuring device. I bivalve, if the number of TLS stations increases, more complete point cloud will be formed.

10- Line 259: "the data quality of the point cloud can not achieve the effect of TLS point cloud."???

Some comments on writing

line 92: In the context of the digital age ???

line 2 124 is duplicated

To use an abbreviation, write the full name in the first instance and follow it immediately by the abbreviated version in parenthesis

line 176: what does it mean: "construct the diameter of ?, and the height of the column for the two point clouds "

line 247: rewrite this sentence "the acquisition method and the point cloud geometry acquisition method are completely different,"

Table 3: compare data ==> compared data

Author Response

Response to reviewer 1 comments

Dear Reviewer,

We deeply appreciate the effort and time you’ve spent in reviewing our manuscript “Feature analysis of scanning point cloud of structure and research on hole repair technology considering space-ground multi-source 3D data acquisition” (ID: sensors-2017741). We studied the comments carefully and have made revisions which we hope meet with your approval. We have modified the manuscript in revision mode. Please find the line number mentioned below in the state of closed revision mode The outline of the revisions is listed as follows:

  • We have had native English speakers check the manuscript for us and make English revisions.
  • We have made changes to the structure of the paper and set the paper according to the following scheme: Introduction, Materials and Methods, Result and discussion section, Conclusions. Through the difference analysis of the multi-source point cloud, the paper selects the TLS point cloud as the reference point cloud according to the difference analysis results, and uses two different source point clouds to achieve the hole repair of the TLS point cloud, so as to optimize the data quality of the TLS point cloud.
  • We redrew the flowchart and explained the content of the flowchart. Meanwhile, according to the new article structure, the positions of the flowchart are adjusted to the line 222 and line240.
  • As for the research motivation you mentioned,we hold the opinion that HLS is to complete the collection of point cloud data in the move and flexibly obtain data from multiple angles. Since the scanning process does not require strict path planning, the position of the scanning head can be actively adjusted in the data collection process. Fur-ther more, the special-shaped structure is prone to data vulnerability areas for key scanning, the integrity of the point cloud can be more guaranteed, and the obtained data can be adopt-ed to check whether there is data missing in the point cloud without any processing. Even though there is a point cloud missing, the location of the missing data can be determined. A new scanning scheme is quickly developed, and data collection is re-performed, so as to complete the missing target point cloud more quickly than TLS. Although TLS designed the acquisition scheme according to the shape of the structure before the experiment, due to the occlusion of the structure itself, the obtained TLS point cloud still lacked data in multiple areas and angles. In order to ensure that the TLS point cloud does not have any data missing within its field of view, it is necessary to repeatedly add multiple stations to collect the point cloud, and match the multi-site cloud data in real time to complete the combination of the structure and the measurement site cloud to ensure the integrity of the point cloud. This operation will not only greatly increase the amount of field operations and reduce the operation efficiency, but also the increase in the number of stations means an increase in the number of matching, and the overall accuracy of the point cloud will also decrease, and the increase in stations will also repeatedly collect point cloud data in the same area, resulting in redundancy of point cloud data. Therefore, simply increasing the number of TLS stations to repair holes to obtain complete data on the structure is not advisable in practice. Please refer to line 309 to line 328.

Comment #1

The abstract is not informative, it should be rewritten..

Response:

Thank you for helpful comment. We rewrote the abstract and introduced the point cloud acquisition technology, the multi-source point cloud difference analysis method and the TLS point cloud hole repair method adopted in the experiment, which enriched the information of the abstract.

Comment #2

the structure of the paper should be mentioned at the end of introduction section.

Response:

Thanks for your this comment. We have described the structure of the paper at the end of the introduction as requested(line109).

Comment #3

The authors stated that: "the spatial resolution of the obtained image is about 1.23cm/pixels.", line 214. However, in line 216 mentioned that "a point cloud is generated at a resolution of 1cm". Is any interpolation applied to raw data?

Response:

Thanks for your helpful comment. We have ignored this problem before, and now it has been explained in line 267 of the article after modification.

Comment #4

Line 216: what the authors mean by "obvious noise"?

Response:

Thank you for helpful comment. We mistakenly used the term "obvious noise" when we meant outliers and structure point clouds.  (line271)

Comment #5

For the HLS point cloud, the authors mentioned that the point cloud is obtained by walking around the object. I am wondering how the top-view is generated in this case.

Response:

Thank you for this comment. "top-view" here refers to the view obtained by projecting all point clouds along the Z-axis into the x and y planes. The previous expression is ambiguous, so the name of the picture has been modified. When we adopt HLS for point cloud collection, the traveling path is about 5m away from the structure. During scanning, the scanning head, driven by the motor, forms a field of view with a 360 degree roll Angle and 270 degree pitch Angle. Therefore, HLS can also obtain the point cloud information of the upper part of the point cloud. We have refined this information on line 293 of the article and in the Schematic diagram of HLS data acquisition path.

Comment #6

line 239: Please explain this: " by removing duplicate points from the data under the condition that the minimum distance between the two points is 0cm.

Response: `

Thank you for this comment. We have revised this statement. Please refer to line 297.

Comment #7

line 249: "average density of the HLS point cloud is 105.45 ??????". Usually the point density is stated in "points per area" or "points per volume".

Response: `

Thank you for this comment. We calculate the volume based on the point cloud density calculation radius, and then change the point cloud density unit.(line 310/508)

Comment #8

line 249: Does having different point densities imply that the local geometry of two point clouds are different.

Response: `

We believe that we have made a mistake. Cloud density at different points does not represent the difference of local geometric features between two points, but only proves that HLS point cloud has a richer amount of data. Therefore, we have revised the original manuscript. Please refer to line311.

Comment #9

Figure 7: Acquiring more texture in HLS is not about the nature of measuring device. I bivalve, if the number of TLS stations increases, more complete point cloud will be formed.

Response: `

Thank you for this comment. TLS structure point cloud has data missing in multiple areas, and in the field collection, it cannot match the data of multiple stations, so these data missing cannot be found in time. If TLS is used to add stations for supplementary measurement of structure, it will be impossible to accurately plan the survey position and point cloud collection range, which will not only cause redundancy of point cloud data, but also reduce operational efficiency. Please refer to line311-line330. In the experiment, the missing area of point cloud from another perspective is selected to supplement this view, as shown in FIG.9, and the data repair of the missing area is completed, as shown in FIG. 26

Comment #10

Line 259: "the data quality of the point cloud can not achieve the effect of TLS point cloud."??.

Response: `

Thank you for this comment. We believe that the cumulative errors in the HLS point cloud will lead to lower coordinate accuracy of the HLS point cloud than that of the TLS point cloud. We have modified the expression in the manuscript and further described this view. Please refer to line334.

Comments on writing

line 92: In the context of the digital age ???

The above expression has been revised in the new manuscript(line120).

line 2 124 is duplicated

The above expression has been revised in the new manuscript.

To use an abbreviation, write the full name in the first instance and follow it immediately by the abbreviated version in parenthesis

We have checked the manuscript for abbreviations and full names

line 176: what does it mean: "construct the diameter of ?, and the height of the column for the two point clouds "

We rewrote the sentence to explain it in more detail.(line202)

line 247: rewrite this sentence "the acquisition method and the point cloud geometry acquisition method are completely different,"

We revised the whole paragraph and also rewrote the sentence

Table 3: compare data ==> compared data

We have corrected this error

Reviewer 2 Report

please find the comments attached. 

1.      The phrases in the paper are too long with many commas (e.g., lines 76-82 is one sentence). This makes the reader confused. The paper should be rephrased.

2.      In Introduction section, more previous studies should be added in the literature review. Only six papers are not enough. Please revise and make search on point cloud data fusion techniques.

3.      Ln 136: it is better to use the term UAS instead of UAV as the UAS includes the UAV plus camera system.

Author Response

Response to reviewer 2 comments

Dear Reviewer,

We deeply appreciate the effort and time you’ve spent in reviewing our manuscript “Feature analysis of scanning point cloud of structure and research on hole repair technology considering space-ground multi-source 3D data acquisition” (ID: sensors-2017741). We studied the comments carefully and have made revisions which we hope meet with your approval.The outline of the revisions is listed as follows:

Comment #1

  1. The phrases in the paper are too long with many commas (e.g., lines 76-82 is one sentence). This makes the reader confused. The paper should be rephrased.

Response:

Thank you for helpful comment. We rephrased the Manuscripts.

Comment #2

  1. In Introduction section, more previous studies should be added in the literature review. Only six papers are not enough. Please revise and make search on point cloud data fusion techniques.

Response:

Thanks for your this comment. In the introduction, we added more previous studies in the literature review as requested.

Comment #3

Ln 136: it is better to use the term UAS instead of UAV as the UAS includes the UAV plus camera system.

Response:

Thanks for your helpful comment. We are required to replace the UAV with UAS where the meaning is the same throughout the Manuscripts

Round 2

Reviewer 1 Report

The quality of the manuscript is improved, significantly. It may be accepted in the current form.